# The Relationship between Indoor and Outdoor Fine Particulate Matter in a High-Rise Building in Chicago Monitored by PurpleAir Sensors

**DOI:** 10.3390/s24082493

**Published:** 2024-04-12

**Authors:** Megan M. Wenner, Anna Ries-Roncalli, Mena C. R. Whalen, Ping Jing

**Affiliations:** 1School of Environmental Sustainability, Loyola University Chicago, Chicago, IL 60660, USA; mwenner@luc.edu (M.M.W.); ariesroncalli@luc.edu (A.R.-R.); 2Department of Mathematics and Statistics, Loyola University Chicago, Chicago, IL 60660, USA; mwhalen3@luc.edu

**Keywords:** fine particulate matter, PM_2.5_, indoor air pollution, Chicago, high-rise buildings, PurpleAir, low-cost sensors, vertical variation

## Abstract

In urban areas like Chicago, daily life extends above ground level due to the prevalence of high-rise buildings where residents and commuters live and work. This study examines the variation in fine particulate matter (PM_2.5_) concentrations across building stories. PM_2.5_ levels were measured using PurpleAir sensors, installed between 8 April and 7 May 2023, on floors one, four, six, and nine of an office building in Chicago. Additionally, data were collected from a public outdoor PurpleAir sensor on the fourteenth floor of a condominium located 800 m away. The results show that outdoor PM_2.5_ concentrations peak at 14 m height, and then decline by 0.11 μg/m^3^ per meter elevation, especially noticeable from midnight to 8 a.m. under stable atmospheric conditions. Indoor PM_2.5_ concentrations increase steadily by 0.02 μg/m^3^ per meter elevation, particularly during peak work hours, likely caused by greater infiltration rates at higher floors. Both outdoor and indoor concentrations peak around noon. We find that indoor and outdoor PM_2.5_ are positively correlated, with indoor levels consistently remaining lower than outside levels. These findings align with previous research suggesting decreasing outdoor air pollution concentrations with increasing height. The study informs decision-making by community members and policymakers regarding air pollution exposure in urban settings.

## 1. Introduction

Nearly 120 million people in the United States experience unhealthy levels of ozone and particulate pollution, and among them over five million are residents or commuters in the city of Chicago [1]. Fine particulate matter (PM_2.5_) is one of the most harmful air pollutants, as these particles are small enough to enter the lungs and some ultrafine particles (<100 nm in diameter) can penetrate the lung walls and enter the bloodstream [2]. PM_2.5_ is also an important topic of study for its relevance to climate change, as a type of aerosol that can have cooling or warming effects on the climate [3]. It is also important for its relevance to environmental justice, as neighborhoods of low-income or high-minority residents tend to experience higher concentrations of and exposure to PM_2.5_ [1,4,5].

In general, PM_2.5_ concentrations in Chicago have decreased in recent years, though the proportion of those concentrations attributable to vehicles and other local sources has increased [6]. Concentrations vary widely across the city and vary significantly by season [7,8]. Additionally, though particulate matter originating from outdoor sources has been widely studied, indoor PM_2.5_ levels have received less attention and are important because outdoor air pollution can infiltrate inside buildings, and indoor sources of particulate matter exist as well. Most Americans spend about 90% of their time indoors [9], so studying indoor PM_2.5_ is important to assess peoples’ overall air pollution exposure.

When examining the variation in PM_2.5_ in an urban setting, elevation is a factor to consider, given the prominence of high-rise apartments and office buildings in which many people live and work daily. Chicago is ranked the second “tallest” city in North America, based on the number of skyscrapers—with 308 buildings in the city over 60 m tall [10]. Furthermore, residents in high-rise buildings tend to be white, young, college-educated, and in higher income brackets [11]. These patterns are a continuation of Chicago’s history of residential segregation. Interestingly, the people living on the highest floors in a given high-rise building are generally those with the highest incomes of all the tenants in the building, with individuals with lower incomes living toward the bottom [12], as units on higher floors offer residents better views and more access to natural light and are therefore more expensive. This is highly relevant to the intersection of environmental justice with the topic of urban air pollution.

Past studies have indicated that air pollutant concentrations tend to decrease with increasing altitude, as demonstrated, for example, by Bisht et al. [13] studying black carbon and particulate matter in Delhi, India; KAILA [14] studying NO_2_ and PM_2.5_ in Helsinki, Finland; and Liu et al. [15] studying PM_2.5_ in two neighborhoods in Nanjing, China. As explained by Liao et al. [16], who studied PM_2.5_ in Taipei, Taiwan, this decrease in many cases may be due to increased distance at higher altitudes from ground-level sources such as traffic emissions. A particular area of concern in urban areas is the problem of street canyons, when buildings lined up on both sides of a road prevent ventilation and dilution of pollutants. According to Liu et al. [15], within street canyons there often exists a specific height at which peak pollutant concentration occurs, which may not be at ground level. How PM_2.5_ varies with height around high-rise buildings in Chicago is not well studied. Therefore, we conducted a project in an office building in the Edgewater neighborhood of Chicago, Illinois.

## 2. Materials and Methods

### 2.1. Sampling Location

To increase our understanding about vertical variations in PM_2.5_ in urban areas, we installed eight PurpleAir sensors (four PA-II outdoor sensors and four PA-I indoor sensors, PurpleAir LLC, Draper, UT, USA) between 8 April and 7 May 2023, in an 11-floor office building in Edgewater, Chicago. The building and the period of the data collection were decided based on the availability of volunteering office occupants who allowed us to temporarily install an indoor sensor inside their office and an outdoor sensor on the exterior wall of their office. All four offices are located on the side of the building that faces away from a major road. Constructed in the late 1950s, the building features traditional brick wall exteriors. In the year 2020, the building was evaluated using the Sustainability Tracking, Assessment and Rating System (STARS), earning an Energy Efficiency Score of 4.66 out of a possible 6.00. This rating denotes that, while the building has adopted several energy-saving measures, opportunities for further enhancements in energy efficiency remain. Throughout the duration of the research period, the building’s windows were kept closed. The building enforces a strict no-smoking policy to support indoor air quality and public health. Additionally, the building is equipped with a centralized cooling/heating system, and heating was operational during the time of the study.

The Edgewater neighborhood, located on the north side of the city, is situated next to Lake Michigan, containing several parks, beaches, and major roadways. According to the Chicago Metropolitan Agency for Planning [17], in 2023 there were about 56,296 residents in Edgewater. The median age is 37.7 years old, the median annual income is $61,872, and the employment rate is 93.5%. Edgewater was a good test neighborhood for its urban location and proximity to roadways and sufficiently tall buildings from which to measure air pollution levels as they vary by height.

### 2.2. PurpleAir Data

PurpleAir is a network of sensors that measure indoor and outdoor PM_2.5_. Data are collected by PurpleAir sensors using a beam from a laser counter that reflects off particles present in an air sample. The reflections translate to estimates of real-time mass concentrations of particulate matter present in the air sample. Each PurpleAir sensor contains two channels, which alternate reading air samples every five seconds, corresponding to an average reading of the sensor every two minutes [18]. We used PA-II outdoor sensors and PurpleAir Touch indoor sensors to monitor outdoor and indoor air quality, respectively. These sensors measure PM_2.5_ and record temperature, relative humidity, and barometric pressure.

PurpleAir sensors were chosen because their data are already integrated into experimental maps (https://fire.airnow.gov (accessed on 16 January 2023)) developed by the U.S. Environmental Protection Agency (EPA) and the U.S. Forest Service. Prior work has evaluated the performance of monitors in the PurpleAir network, and newly established correction factors can be used to correct reported PM_2.5_ values to better agree with Federal Equivalent Method (FEM) measurements [19,20,21,22]. Following the recommendation for outdoor air quality monitors in the U.S., as used in O’Dell et al. [23], we used the correction factor developed by Barkjohn et al. [22], as shown below:

For PA_cf_1_ < 343 μg/m^3^:PM_2.5_ = 0.524 × PA_cf_1_ − 0.0862 × RH + 5.75 (1)
and for PA_cf_1_ ≥ 343 μg/m^3^: PM_2.5_ = 0.46 × PA_cf_1_ + 0.000393 PA_cf_1_^2^ + 2.97(2)
in which PA_cf_1_ is the average of the PM_2.5_ values in μg/m^3^ measured by Channels A and B over a two-minute interval with the conversion factor equal to 1, and RH is relative humidity in percent. According to O’Dell et al. [23], this correction factor can be applied to indoor sensors as well, and therefore we applied it to measurements taken by both outdoor and indoor sensors. In addition, we discarded data points whose PM_2.5_ values were less than 0 μg/m^3^ or greater than 80 μg/m^3^. According to more reliable data retrieved from EPA air quality monitors, in this location there were no days during our study period in which the average PM_2.5_ levels exceeded 80 μg/m^3^. Concerns have been raised about the efficacy of correction factors under conditions of elevated humidity. Throughout the duration of the present study, recorded outdoor RH levels remained below 60% and indoor RH levels never exceeded 20%. Consequently, the potential influence of high RH on the applied correction factors is considered to be negligible.

The efficacy of PurpleAir sensors is influenced by variables such as particle size, composition, and the origin of particulates, all of which vary between indoor and outdoor settings. The use of the correction factor intended for outdoor sensor data when analyzing readings from indoor sensors introduces a potential source of inaccuracy in our analysis. We recognize this is a limitation of our study.

### 2.3. The Correlation between Outdoor and Indoor PM_2.5_

We analyzed the correlation between outdoor and indoor PM_2.5_ to examine the infiltration of outside air into the building following the method of Lv et al. [24], expressed with the following equation:C_in_ = F_in_C_out_ + C_s_(3)
where C_in_ is the actual indoor PM_2.5_ concentration, C_out_ is the actual outdoor PM_2.5_ concentration, and C_s_ represents the concentration of PM_2.5_ generated from indoor sources. F_in_ is the infiltration coefficient, representing the factor by which PM_2.5_ generated outdoors enters into the building, which depends on the ventilation system and permeability of the building. In the most efficiently ventilated buildings, F_in_ should theoretically be zero, indicating that the building is not “leaky” or vulnerable to pollutant infiltration from the outdoors.

### 2.4. Colocation and Data Collection

We co-located our four outdoor sensors by placing them side-by-side together outside the office building for a week between 29 March and 4 April 2023, to determine to what extent the sensors’ readings agreed with each other. To correct the differences observed between sensors during this co-location, we used these data to calculate how much the mean PM_2.5_ concentration of each sensor deviated from the mean across all four sensors. These amounts of deviation were added to all future data points for each respective sensor. The same process was conducted for the four indoor PurpleAir sensors co-located inside over the same week-long period. 

We then installed a pair of indoor and outdoor sensors for an office on each floor (one, four, six, and nine) of the building (Table 1). They were positioned on the inside and outside of the office’s exterior wall that faces away from the roadside and away from the lake. The four offices were selected based on the voluntary willingness of occupants to allow installation for one month, i.e., between 8 April and 7 May 2023. In addition to the data provided by the sensors we installed, we obtained data from another outdoor PurpleAir sensor in Edgewater, situated on the fourteenth floor of a condominium about 800 m away from the office building. Even though this sensor was on a different building, we decided it would be beneficial to examine its data as well because of the higher elevation of the sensor.

## 3. Results

We conducted co-location tests for both indoor and outdoor sensors for a week between 29 March and 4 April 2023. Raw PM_2_._5_ measurements taken by each sensor were corrected based on the relative humidity values using Equations (1) and (2). We then interpolated the corrected PM_2.5_ to regularly gridded two-minute intervals to perform a comparative evaluation of the sensors. Our results show that each outdoor sensor agreed closely with the other outdoor sensors (the R-squared value between any two sensors is between 0.97 and 0.99) (Figure 1). The same was true for the co-location results of the indoor sensors (the R-squared value between any two sensors is between 0.96 and 0.97) (Figure 2). The average PM_2.5_ concentration of each outdoor sensor during the week displayed biases that ranged between −0.16 and +0.15 μg/m^3^ from the mean value of all the four outdoor sensors combined. The biases for the four indoor sensors ranged between −0.10 and +0.22 μg/m^3^. These biases were removed from the observations before we compared PM_2.5_ levels at different floors in the office building. The co-location test did not include the public outdoor sensor installed on the 14th floor of the condominium. Therefore, we did not remove any potential bias for this 14th floor sensor.

During the one-month study period between 8 April and 7 May 2023, data collected by the outdoor sensors were first corrected based on the relative humidity value using Equations (1) and (2) and then their biases were removed based on the co-location results. The results indicated that the median PM_2.5_ concentration increased with increasing height between the 1st and 4th floors (Figure 3). The median outdoor PM_2.5_ concentrations decreased for each successive level of altitude beyond the 4th floor by 0.11 μg/m^3^ per meter elevation. Statistical tests (Kruskal–Wallis test and Dunn test) showed significant differences between every outdoor sensor compared to every other outdoor sensor, except for one pair (the 1st and 9th floors). For indoor air quality, the median PM_2.5_ concentration increased with increasing height (+0.02 μg/m^3^ per meter elevation) for all levels (Figure 4). Although small, these differences between every indoor sensor were statistically significant.

We calculated the hourly average PM_2.5_ concentrations to study the diurnal cycles of the indoor and outdoor PM_2.5_ at different floors. Outdoor PM_2.5_ concentrations displayed a similar daily cycle at all floor heights (Figure 5). They all decreased in the afternoon until 8 p.m., and then increased throughout the night and early morning followed by a drop between 9 a.m. and 10 a.m. and increased again until peaking at noon. Such variations are controlled not only by the diurnal changes in PM_2.5_ emissions from traffic but also by the dynamics of the atmospheric boundary layer [25]. The vertical mixing in the boundary layer is weak at night and strongest in the mid-afternoon, causing air pollutants to accumulate at night and disperse in the afternoon. The most pronounced differences of PM_2.5_ by floor height occurred between 12 a.m. and 8 a.m. when the atmospheric condition is stable and vertical mixing is weak. Figure 5 also shows that PM_2.5_ concentrations decreased as the height of the sensor increased except for the 1st floor. The fact that the 1st floor PM_2.5_ concentrations are lower than those at the 4th floor is because the sensor at the 1st floor is situated above bushes and surrounded by trees, which could have helped to clean the air by removing PM_2.5_ from the air. Indoor PM_2.5_ concentrations, however, remained relatively constant over a day (Figure 6), though with a late-morning spike in the concentration on the 9th floor. The greatest variation in indoor PM_2.5_ by floor height appeared between the hours of 9 a.m. and 3 p.m., likely caused by the activities of office occupants during their work hours.

To understand what caused the indoor air quality to change, we compared the indoor and outdoor daily average PM_2.5_ concentrations observed by the four paired indoor and outdoor sensors. By conducting a linear regression test, our results show a positive correlation between the indoor PM_2.5_ and the outdoor PM_2.5_ (Figure 7 and Table 2), indicating the influence of outside air on the indoor air quality through infiltration. The infiltration factors, represented by the slope of the linear fitted line (F_in_), vary between 0.22 and 0.29. The intercept of the linear fitted line represents the average amount of PM_2.5_ attributable to indoor sources (C_s_), and it ranges between 3.91 μg/m^3^ and 4.23 μg/m^3^ across the four floors. In comparison, the C_s_ and F_in_ values for office buildings in Daqing, China, were 4.83 μg/m^3^ and 0.72, respectively [24]. We then estimated the contribution of outdoor PM_2.5_ to indoor PM_2.5_ using the method in Lv et al. [24] by calculating C_out_F_in_/C_in_. Our results show the contributions of outdoor PM_2.5_ to the mean indoor PM_2.5_ are between 30% and 40% across the four floors, which are lower than the contribution rate of 79% in Lv et al. [24].

## 4. Discussion

This study conducted in Edgewater, Chicago, focused on analyzing the vertical profile of PM_2.5_ concentrations in both outdoor and indoor environments of high-rise buildings using PurpleAir sensors. The near-ground layer of the atmosphere is typically not probed by conventional instruments such as satellites, lidar, or air balloons. The PurpleAir sensors offer a low-cost method to investigate the fine vertical resolution of PM_2.5_ concentration change in the area.

We found good agreement among PurpleAir sensors during the one-week co-location period between 29 March and 4 April 2023, indicating the reliability of the sensors for the study. The findings revealed that PM_2.5_ concentrations decreased at 0.11 μg/m^3^ per meter for each successive level beyond the 4th floor. This trend may be attributed to the increasing distance from ground-level pollution sources, such as traffic, as elevation increased. The fact that PM_2.5_ on the 1st floor is lower than on the 4th floor is likely due to the fact that the outdoor sensor was situated above bushes and there are trees nearby. The vegetation could have helped to increase the dry deposition of PM_2.5_. Indoor PM_2.5_ concentrations increased with increasing height at 0.02 μg/m^3^ per meter elevation, from 5.3 μg/m^3^ at the first floor to 5.8 μg/m^3^ at the 9th floor, which is mostly likely caused by the greater infiltration coefficient, which is 0.22 at the 1st floor and 0.29 at the 9th floor.

Hourly averages of outdoor PM_2.5_ concentrations showed the clearest variation by height between the hours of 12 a.m. and 8 a.m. The same pattern of outdoor PM_2.5_ concentration increasing between the 1st and 4th floors, then decreasing after the 4th floor, still holds over this period. Mornings exhibit the most variation due to the low atmospheric boundary layer overnight, resulting in little atmospheric mixing and more distinct stratification with height. For the indoor sensors, the clearest variation in air quality by height appeared to be between 9 a.m. and 3 p.m. This could be due to this period being the peak of the workday, with the most activity in the building occurring then.

Comparing outdoor to indoor PM_2.5_, for each floor their concentrations were positively correlated; the indoor air pollution stayed at a concentration lower than outside. This follows logically given that we expect most of the PM_2.5_ sources in this setting to originate outdoors. The F_in_ values, or the slopes of the regression lines, are low and indicate that there are low factors of infiltration of outdoor PM_2.5_ penetrating inside the building. The C_s_ values, or the indoor intercepts of the regression lines, indicated that the sources of indoor air pollution were low, around 4 μg/m^3^.

A potential limitation of this study was that the sensors at different floor heights were not placed exactly above one another, but distributed along the same wall of the building according to the location of the office in which an occupant was willing to host a sensor. Additionally, the chosen condominium building is a few blocks away and may experience slightly different primary emission sources of PM_2.5_ compared to the office building. In our future research on indoor air quality, sensor installation will adhere to the guidelines established by the American Society of Heating, Refrigerating, and Air-Conditioning Engineers (ASHRAE).

Our examination of the relationship between indoor and outdoor PM_2.5_ concentrations is constrained by insufficient data regarding indoor environmental variables, including air infiltration rates and the presence of particles originating from within the building. Additionally, a lack of detailed information on outdoor meteorological factors, such as wind patterns and precipitation, further limits the comprehensiveness of our analysis.

This project serves as a case study for two buildings in Edgewater, Chicago, for one month in spring. A full understanding of the vertical profile of PM_2.5_ concentration requires a longer and more extensive period of study, incorporating greater temporal and spatial coverage and greater vertical extent. Future research could include a greater number of different sensor heights and could incorporate a greater degree of control between heights, with data collection periods spanning multiple seasons of the year and examining how weather impacts outdoor air quality and how building type influences infiltration indoors.

Research on this topic will provide a greater nuance to our understanding of how people in urban areas may be exposed to different levels of air pollution while on different building floors, as air quality is an important topic for its relevance to climate science, public health, and environmental justice.

## Figures and Tables

**Figure 1 sensors-24-02493-f001:**
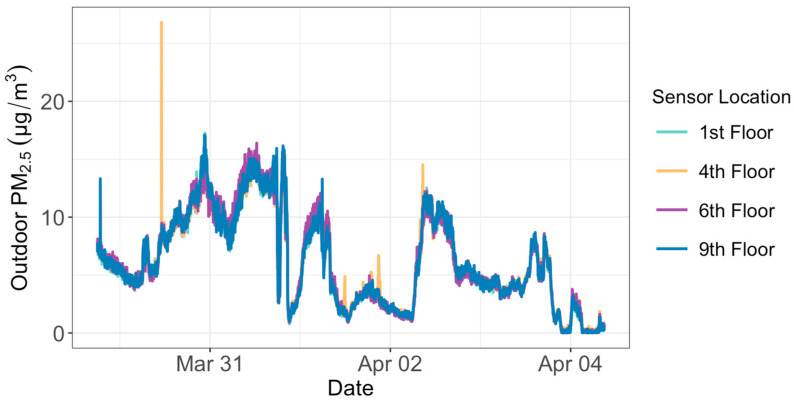
PM_2.5_ concentrations of the four outdoor PurpleAir sensors when they were co-located between 29 March and 4 April 2023, before they were later installed on the outside walls of the four floors of an office building.

**Figure 2 sensors-24-02493-f002:**
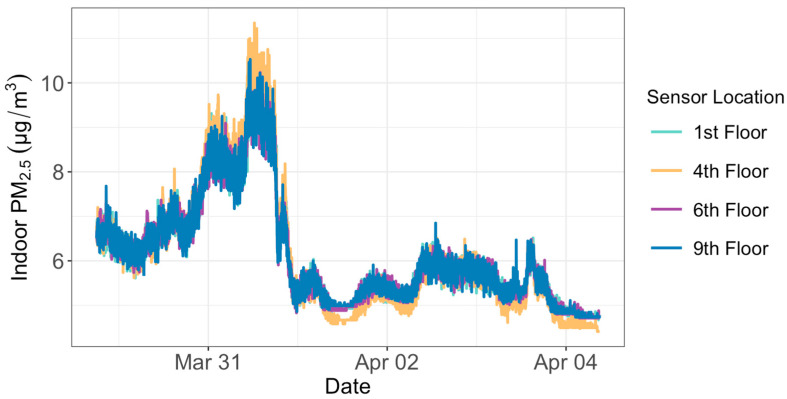
PM_2.5_ concentrations of the four indoor PurpleAir sensors when they were co-located between 29 March and 4 April 2023, before they were later installed inside offices on the four floors of an office building.

**Figure 3 sensors-24-02493-f003:**
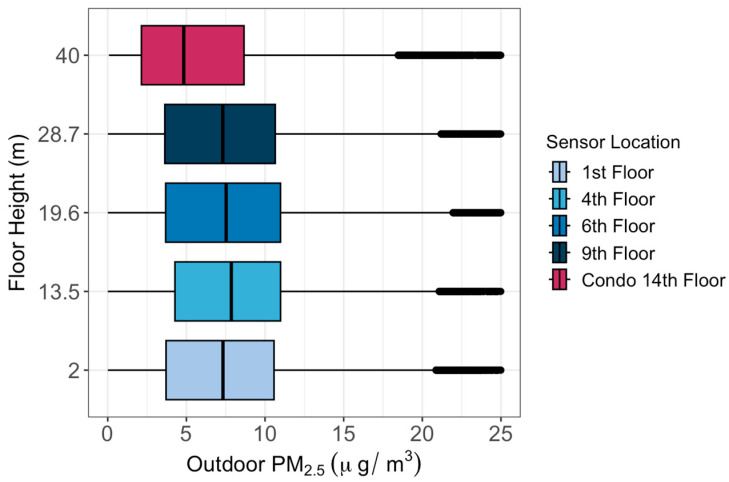
Distributions of outdoor PM_2.5_ concentrations at different floor heights between 8 April and 7 May 2023.

**Figure 4 sensors-24-02493-f004:**
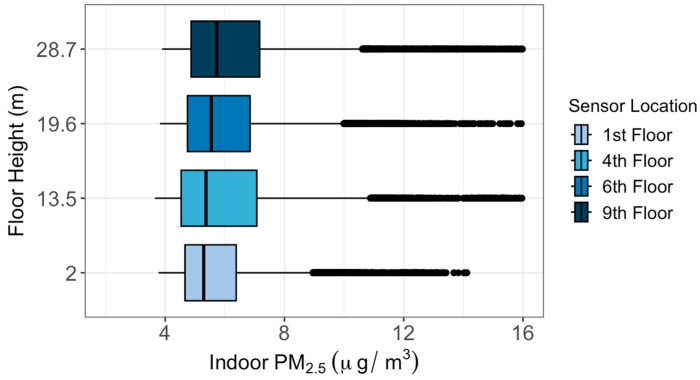
Distributions of indoor PM_2.5_ concentrations at different floor heights between 8 April and 7 May 2023.

**Figure 5 sensors-24-02493-f005:**
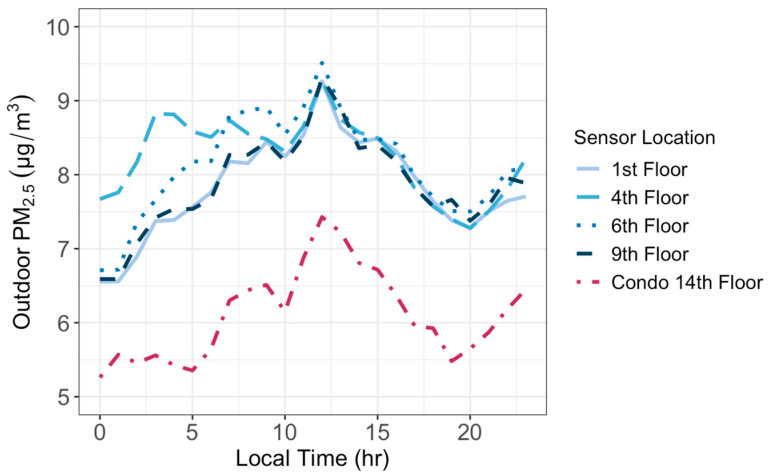
The average hourly PM_2.5_ concentrations observed by outdoor sensors at five different floor heights between 8 April and 7 May 2023.

**Figure 6 sensors-24-02493-f006:**
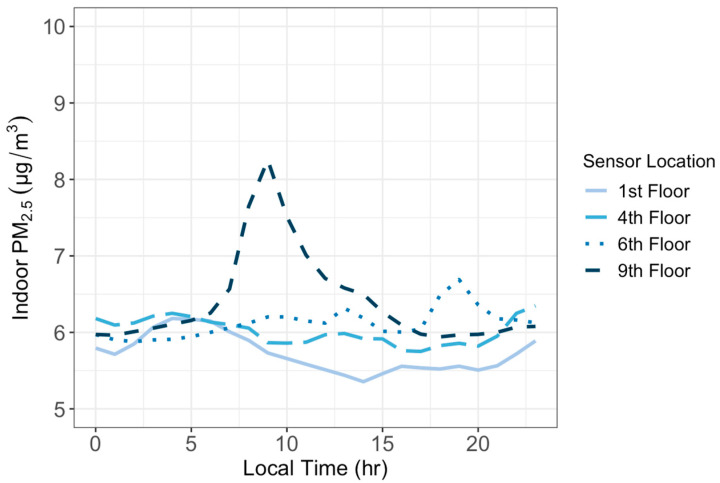
The average hourly PM_2.5_ concentrations over a 24 h day observed by the indoor sensors at four floor heights between 8 April and 7 May 2023.

**Figure 7 sensors-24-02493-f007:**
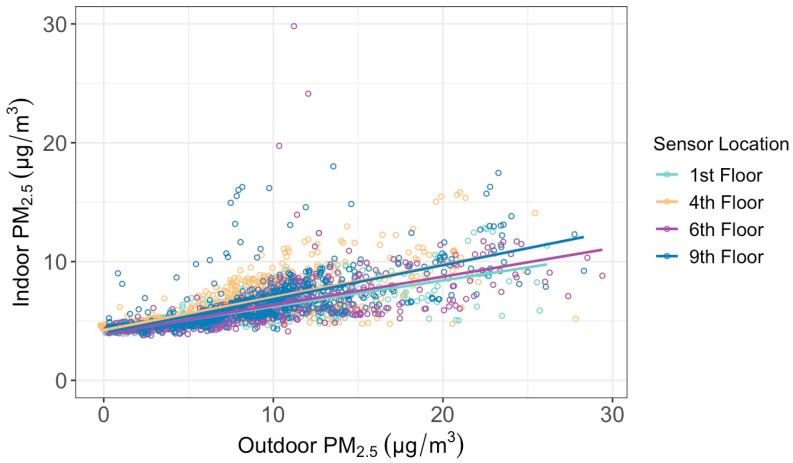
Indoor versus outdoor daily average concentrations of PM_2.5_ at four floor heights between 8 April and 7 May 2023.

**Table 1 sensors-24-02493-t001:** The information of the sensors used in this study, including their PurpleAir sensor identification number, location, elevation, latitude, and longitude.

Sensor IDs	Location	Elevation (m)	Latitude	Longitude
175445 (outdoor); 171081 (indoor)	Office Building 1st Floor	2.0	41.9979	−87.6566
175417 (outdoor); 171015 (indoor)	Office Building 4th Floor	13.5	41.9980	−87.6567
175457 (outdoor); 171075 (indoor)	Office Building 6th Floor	19.6	41.9979	−87.6563
175419 (outdoor); 171079 (indoor)	Office Building 9th Floor	28.7	41.9980	−87.6564
123003 (outdoor)	Condominium 14th Floor	40	41.9911	−87.6546

**Table 2 sensors-24-02493-t002:** Results of the linear regression model between indoor (C_in_) and outdoor (C_out_) PM_2.5_, C_in_ = F_in_C_out_ + C_s_.

Sensor Location	Indoor Median PM_2.5_	Outdoor Median PM_2.5_	Intercept (C_s_)	Slope (F_in_)	R-Squared	*p*-Value
1st Floor	5.29 μg/m^3^	7.36 μg/m^3^	4.0197 μg/m^3^	0.2196	0.6125	<2.2 × 10^−16^
4th Floor	5.37 μg/m^3^	7.90 μg/m^3^	4.2349 μg/m^3^	0.2762	0.5261	<2.2 × 10^−16^
6th Floor	5.55 μg/m^3^	7.59 μg/m^3^	3.9101 μg/m^3^	0.2411	0.4346	<2.2 × 10^−16^
9th Floor	5.75 μg/m^3^	7.37 μg/m^3^	3.9936 μg/m^3^	0.2853	0.5059	<2.2 × 10^−16^

## Data Availability

All raw data are publicly available from the PurpleAir website (https://www2.purpleair.com/ (accessed on 15 May 2023)) using the PurpleAir API (https://api.purpleair.com/ (accessed on 15 May 2023)). All analyzed data available on request from the corresponding author.

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
