# Peer review of "The Relationship between Indoor and Outdoor Fine Particulate Matter in a High-Rise Building in Chicago Monitored by PurpleAir Sensors"

_sensors, 2024, doi:10.3390/s24082493_

Round 1

Reviewer 1 Report

Comments and Suggestions for Authors

The manuscript brings important data from a case study of 2 buildings in Edgewater, Chicago, using low-cost sensors from PurpleAir and measurements of PM 2.5, in 4 floors, indoor and outdoor, plus one set of data from a nearby higher building.

As growing populations around the world live in megacities and tend to work and/or live in tall buildings, the work is relevant for other world contexts. With this in mind, a believe that an improved description of the building's architectural characteristics and surrounding is necessary. Is the building closed to the outside, or is it possible to open windows? What are the conditions of the air conditioning system and care for it? Is smoking allowed?

How does Edgewater neighborhood differ from other Chicago neighborhoods? A map could be interesting to understand the geographical landscape. It is mentioned that air quality was better on the lower level and the explanation given was the existence of vegetation. This could be further explored in the discussion.

What is the point in giving percentages of racial composition in Edgewater? In the introduction the issue of environmental justice is mentioned, but data collected, analysis, and discussion do not deal with this. So, what is the point the authors want to make?

In the methods section, it is informed that the samplers collect data on PM 2.5, temperature, RU, and barometric pressure. Why this information was discarded and only particulate matter was analyzed? I also suggest that information on predominant wind speed and direction, and precipitation during the month of the study be included in the discussion. It was good that authors pointed the limitation of the season of the year (spring) and the reduced number of measuring points in the research. Were there other limitations that could be mentioned?

Reviewer 2 Report

Comments and Suggestions for Authors

The manuscript “The Relationship between Indoor and Outdoor PM2.5 in a High-Rise Building in Chicago Monitored by PurpleAir Sensors” describes a study conducted in Chicago where a number of PM sensors were installed inside and outside a high-rise building and the PM data were measured by height and compared to each other (indoor and outdoor) and to other similar studies.

 The paper is well written and properly organized. The methods and results are clear and the discussion includes all relevant data.

 As a note to the authors, however, it seems that your paper could benefit from a wider range of data (more sensors / altitude difference / seasons, etc., etc.) and a more detailed discussion with similar studies.

 Nevertheless, the manuscript provides sufficient data and, in my opinion, can be published as is.

Reviewer 3 Report

Comments and Suggestions for Authors

A simple request: please use multiple colors rather than variants of blue for figures 1 and 2. Thank you.

There are certain statements that need correcting, clarification or caveats:

1  Line 31: your reference states that some particles can enter the bloodstream. Toxicology studies show that ultrafine particles (<100 nm dia) are the commonly accepted diameter for penetrating the lung walls- this is below the lower detection limit of the optical based Plantowers (about 300 nm dia). You should rewrite the sentence, including that not all PM2.5 particles enter the bloodstream, only the UFPs.

2  The referenced EPA paper from which you used the correction equation: they agreed that they had very little high RH data; their linear RH correction is not consistent with hygroscopic growth factors which are exponential. Your indoor data is possibly OK because the effect of particle growth is nominal below 55%RH, but your outdoor readings will be affected. I wish you had plotted the RH of the outdoor sensors, although the Plantowers have a large negative bias error on their RH readings due to internal heating of the fan and laser.

3  Minor point, but the constants used by the US EPA are 0.0862 and 0.524: three significant digits, not two.

4  You seem to have excellent agreement between the two channels in the Plantowers, yet the EPA paper had significant problems with lack of agreement (5ug/m3 or 61% difference ) between the two Plantower sensors in their 50 Purple Airs. Can you comment?

5  You reference O'Dell to align indoor and outdoor particle measurements. But O'Dell referred to wildfires where the particle size distribution is different than in an office environment. Also, the mean of the wildfire measurements was between 60 and 80 ug/m3, much greater concentrations than you encountered.  This reference should not be used to support your assumption.

6  I liked the use of equation 3 for infiltration calculations. You missed a trick: the Plantowers apparently also report pressure, so you could have studied delta P vs delta PM. Delta P is the driver for infiltration.  I also suspect that some of your first floor irregularities could be due to pressure differences/ local airflows at ground level. You could also have looked at vertical pressure difference when you had large variances between the floors.

10  When monitoring spaces you should define the activities in each space and there are ASHRAE rules for locating PM monitors in spaces: the distance from walls and height from the floor are both critical. See ASHRAE 62.

11 The location seems to be facing the lake- if that is correct, then many of the outdoor generated particles will be a combination of BC/soot and NaCl from the lake. NaCl particles are very hygroscopic and calibrating for RH requires a different coefficient- see my comment above.

12 Figure 5: I was expecting either daily traces or error bars, not a single trace. Was this the average of the week or just a day that you selected?

13  I am not sure of the use of the height vs PM concentration constant- it will depend on the met conditions, particle type, local airflows, infiltration rates and rate of internally generated particles.

Round 2

Reviewer 3 Report

Comments and Suggestions for Authors

The manuscript is improved. Thank you.

You should have used different colors rather than shades of blue in figure 7. 

Not facing the lake does not avoid NaCl aerosols.

I am surprised that indoor RH is below 20%RH- that is very dry and uncomfortable.

Author Response

We thank the reviewer for proving the constructive feedback to our revised manuscript.

We have updated Figure 7 according to the reviewer's suggestion. Please see the Manuscript Revision 2.

We appreciate the reviewer's insightful comment about NaCl aerosols. It will be very interesting to investigate the composition of PM2.5 at different altitudes in Chicago. We are excited about the possibility of exploring this further in our next study.

The indoor RH was uncomfortably low in that office building, likely due to the fact that the heating that was still on. We plan to communicate our research findings with the building's occupants and management, highlighting the potential impact on comfort and health.

Once again, we would like to extend our sincere thanks to the reviewer. Your valuable comments have not only enhanced the quality of our current manuscript but have also sparked new ideas for our next research project on Chicago's indoor air quality.  Your comments have been instrumental in guiding our research direction and improving our work.